# Comparison of time to death and its predictors among hospitalized children with and without severe acute malnutrition at Mulanje District Hospital, Southern Malawi

**Masuzgo Rodney Pajogo**[1]*, **Miriam Ndholvu**[1]

1 Department of Public Health, Kamuzu University of Health Sciences, Blantyre, Malawi,

* masuzgopajogo@gmail.com

## Abstract

### Introduction

Severe acute malnutrition (SAM) is a major cause of child mortality in sub-Saharan Africa, yet recent data from Mulanje District Hospital, Malawi, showed higher mortality among non-SAM under-five children. This unexpected trend highlights a knowledge gap, as no studies in Malawi have compared time to death and its predictors between SAM and non-SAM children. This study aimed to fill that gap by examining and comparing mortality timing and predictors in both groups.

### Methods

A retrospective cohort study was conducted using medical records of 454 randomly selected under-five children admitted to Mulanje District Hospital between January 2017 and February 2021. Data were collected using structured forms and analysed in STATA version 16. Cox proportional hazards regression was used to identify mortality predictors, with significance set at $p < 0.05$.

### Results

The overall mortality rate was 14.8%, with higher mortality in non-SAM children (21.2%) than SAM children (8.4%). The median time to death was 5 days (IQR: 2–8) for SAM and 1 day (IQR: 1–2) for non-SAM children. Among SAM children, not having received amoxicillin (AHR: 4.59; CI: 1.46–14.43) was a significant predictor of death. Among non-SAM children, facility referral (AHR: 2.66 (95% CI: 1.34–5.27)), oxygen therapy (AHR: 4.04 (95% CI: 2.11–7.71)), and not having received amoxicillin (AHR: 33.49 (95% CI: 4.47–250.7)) were significant predictors of mortality.

which permits unrestricted use, distribution, and reproduction in any medium, provided the original author and source are credited.

**Data availability statement:** Data cannot be shared publicly because it contains potentially identifying information. Data are available from the Mulanje District Hospital's Research Coordinating Committee representative, Lloyd Mathewe, via email (llodymathewe23@gmail. com) or via telephone ((+265)888117349), for researchers who meet the criteria for access to confidential data.

**Funding:** The author(s) received no specific funding for this work.

**Competing interests:** The authors have declared that no competing interests exist.

**Abbreviations:** ARI, Acute Respiratory Infection; CFR, Case Fatality Rate; COMREC, College of Medicine Research and Ethics Committee; CMAM, Community-based Management of Acute Malnutrition; HIV, Human Immunodeficiency Virus; IV, Intravenous Fluids; LMIC, Low-and middle-income countries; MoH, Ministry of Health; MSF, Médecins Sans Frontières; NGT, Nasogastric tube; NRU, Nutrition Rehabilitation Unit; ORS, Oral Rehydration Solution; ReSoMal, Rehydration Solution for Acute Malnutrition; RUTF, Ready to use therapeutic feeds; SAM, Severe Acute Malnutrition; SSA, Sub-Saharan Africa; TB, Tuberculosis; WHO, World Health Organization.

## Conclusion

The higher mortality observed among non-SAM children reflected more acute disease presentations and delays in effective intervention, underscoring the need for rapid triage and treatment in this group.

## Background

Severe acute malnutrition (SAM) is defined by a very low weight-for-height (below a −3 z-score of the median World Health Organization (WHO) growth standards), visible wasting, or the presence of nutritional oedema [1]. Typically, SAM manifests itself in two forms: marasmus, characterized by severe wasting, and kwashiorkor, characterized by bilateral pitting oedema [2]. Both marasmus and kwashiorkor are associated with several co-morbidities, and they contribute significantly to high rates of hospital admission and mortality in childhood [3].

Globally, SAM remains a public health problem. Every year, almost one million children under five years of age die because of SAM, accounting for nearly 12–15% of all under-five deaths worldwide [4]. Global reports have shown that 45% of child deaths in developing countries are attributed to SAM [5]. In 2018, nearly 17 million children under five years old had SAM in 2018, with 2.9 million receiving inpatient treatment. Of these cases, 4.4 million were cases from sub-Saharan Africa [6]. By 2020, 6.7% of the global under-five population (45.5 million children) experienced at least moderate wasting, including 13.6 million who were severely wasted. Evidence shows that two-thirds of all wasted children live in Asia and over one-quarter in Africa [5]. Meanwhile, the death rate of children under 5 years of age remains between 10 and 40% in sub-Saharan Africa, including Malawi [7,8].

Under-five mortality is a significant public health concern in Malawi [9]. According to the United Nations Children's Fund (UNICEF) Malawi, approximately 40,000 children under five die annually from preventable or treatable conditions. Leading causes of under-five mortality in Malawi include pneumonia (14%), diarrhoea (8%), and malaria (7%), with SAM contributing to nearly 25% of these deaths [10].

Severe acute malnutrition is a major contributor of under-five mortality and morbidity in Malawi. Nationally, 37% of children under five years of age are stunted, 3% are wasted, and 12% are underweight [11]. At Queen Elizabeth Central Hospital in Blantyre, SAM-related mortality in under-five children has reached up to 42% [12]. In 2015, the Malawi's national case fatality rate (CFR) for SAM in was 9.6%. However, seven hospitals in Malawi, including Mulanje District Hospital, had an average CFR of 11.6%, exceeding the Sphere standard of <10% [13]. In Malawi, the analysis of Community Management of Acute Malnutrition (CMAM) trends showed fluctuating rates, with the CFR of 7.9% in 2016, 9.7% in 2017, 8.5% in 2018, and 11% in 2019 [14].In view of this, researchers in Malawi have conducted several studies to investigate predictors of mortality in children with SAM [15–17]. Studies have identified human immunodeficiency virus (HIV) infection, Kwashiorkor, shock, and acute respiratory infections (ARI) as independent risk factors for the high death rate among children with SAM [15].

Generally, severely malnourished children do not respond to treatment in the same way as well-nourished children. Therefore, the management of SAM children differs from that of non-SAM children [18]. Non-SAM children are managed by treating the diagnosed condition using appropriate medications. The management of SAM involves several parameters, including nutritional therapy, routine medical treatment, and the management of common complications. Nutritional therapy involves the use of therapeutic feeds enriched with vitamins and minerals. The WHO guidelines recommend the use of Formula 75 (F-75), Formula 100 (F-100), and RUTF in the management of SAM [19]. The medical treatment uses broad-spectrum antibiotics regardless of signs of clinical infections. This is because SAM children may not show signs of infection due to reductive adaptation. As recommended by the WHO, children with uncomplicated SAM are treated with oral broad-spectrum antibiotics, whereas children with complicated SAM are treated with intravenous (IV) broad-spectrum antibiotics followed by oral antibiotics [20]. Meanwhile, evidence shows that the use of antibiotics improves growth and reduces the mortality of children with SAM [21].

The management of medical complications in children with SAM depends on the type of complication affecting the child. The most common complications affecting children with SAM include diarrhoea, dehydration, shock, bacterial infections, hypothermia, and hypoglycemia [22]. According to WHO treatment guidelines for SAM, diarrhoea or dehydration is managed with ReSoMal. Bacterial infections, such as lower respiratory tract infections and urinary tract infections, are treated with broad-spectrum antibiotics. Hypothermia is managed by covering sick children with blankets as well as keeping the child closer to the mother's body. Hypoglycemia is managed with an intravenous infusion of 10% glucose, IV antibiotics and F-75 [19,20,22]. Meanwhile, the WHO guidelines have proven to be effective, with a CFR of less than 5% even in resource-constrained settings when applied with diligence [23].

Mulanje District Hospital is one of the district hospitals in Malawi that is heavily affected by SAM. On average, each year, the hospital admits 1000–4000 SAM children [24]. A substantial proportion of the malnourished children die while receiving medical treatment. A report has also shown that the mortality rate among children admitted with SAM in the district had risen from 4.3% in 2014 to 8.4% in 2015 [13]. While it is well known that mortality rates among children with SAM remain high in the district, no study has been conducted to compare the time to death and its predictors between SAM and non-SAM children. Therefore, this study was aimed at comparing the time to death and predictors of mortality between SAM and non-SAM under-five children admitted at Mulanje District Hospital. This study addressed a gap in the extant literature by determining how time to death and predictors of mortality differ between SAM and non-SAM children. By comparing SAM and non-SAM groups, this study aimed to uncover clinically actionable risk factors across a spectrum of nutritional statuses, enhancing triage and intervention strategies for in-hospital paediatric care.

## Methods and materials

### Study area

The study was conducted in Mulanje district, a semi-urban district in the southern region of Malawi, Africa. Mulanje district covers 2056 km$^2$, and has a total population of 684,107, with 23 health centres, 1 mission hospital, and 1 district hospital [25]. Mulanje District Hospital has over 500 beds, of which 100 are in the paediatric ward. Ninety-nine beds are for non-SAM children, while 10 beds are set aside to serve patients suffering from SAM.

### Study design and study period

This was a retrospective cohort study that was conducted from February 2021 to June 2022.

### Study population

The study population consisted of children under five years of age admitted to Mulanje District Hospital between January 2017 and February 2021.

## Inclusion criteria

- Non-SAM and SAM patients aged 0–59 months

- All children admitted with medical conditions.

## Exclusion criterion

- Admitted children aged 5 years and above.

- All hospitalised children who were not assessed for their nutrition status

- Admitted children with surgical and orthopaedic conditions. These children were excluded due to their distinct patho-physiological profiles and clinical pathways. Their outcomes are more influenced by surgical interventions and injury severity than by nutritional or infectious factors. Including them could have introduced heterogeneity and confounded the relationship between nutritional status and mortality

## Sample size determination

The sample size was determined using the formula for detecting a difference between two proportions. We considered the following assumptions to determine the appropriate sample size: an 11.6% mortality rate in the exposed group [7], a 4.16% mortality rate in the unexposed group [26], a 95% confidence interval, an 80% power, and a 10% contingency. After calculations, the total sample size was 412. After adjusting for 10% of the missing files, the final sample size was 454. The sample size among the exposed (n1 = 227) and the sample size among the unexposed (n2 = 227).

## Data collection

We used structured data collection forms to collect data from the medical records of the children under 5 years of age who were admitted from January 2017 to February 2021. Patients' files were collected from the storage room in paediatric department. Data for SAM children was collected from the NRU. Data for non-SAM children were collected from the paediatric medical ward. We developed data collection forms by following the format of the present paediatric admission sheet at Mulanje District Hospital [27]. The parameters in the data collection form followed the order in which information appears in the patient's records. We designed the first section of the form to collect socio-demographic data. The subsequent sections collected data on clinical characteristics, comorbidities, and routine treatment. The last section collected data on treatment outcomes.

We conducted a pilot test on the data collection form and made appropriate changes as needed. During data collection, medical charts were followed up for 30 days post-admission to assess the occurrence of death. Two nurses were recruited to collect data from the medical records. Data were collected by reviewing patients' medical charts. Multiple imputation was used to handle the missing data. Missing values were replaced using predictive models based on other variables in the dataset. We collected the following information from the medical charts:

1. Social-demographic data: This included age, sex, referral status, and date of admission.

2. Clinical data: This included fevers, vomiting, diarrhoea, dehydration, shock, anaemia, coughing, convulsions, and oedema.

3. Routine treatment data: This included administered medications, oxygen therapy, and nasal-gastric (NG) feeding tube.

4. Treatment outcome: This included information about treatment outcomes, including whether the patient had died or was censored.

## Sampling procedure

Annually, the paediatric department at Mulanje District Hospital admits approximately 3000 children. A total of 7,685 children under five years of age were admitted between January 2017 and February 2021. Of these, we drew 643 samples from NRU and 5614 samples from the paediatric medical ward, which fulfilled the inclusion criteria. 1428 patients did not fulfil inclusion criteria, and therefore their charts were excluded from the study. All SAM and non-SAM under-five children who had met the inclusion criteria were separately listed in cells in the Excel spreadsheet. We used the RAND function to assign a random number to each cell and selected the required sample size of 454 from the eligible patient population using an index rank formula.

## Data quality control

We achieved data quality by creating a data collection form that was appropriate for the study's objectives. Besides this, we used two qualified nurses for data collection. Before data collection process, the nurses underwent a two-day intensive training that covered the study protocol, operational definitions of variables, and best practices for extracting data from medical records. The training aimed to enhance their competence, ensure consistency in data collection, and improve efficiency.

To assess and reinforce consistency between data collectors, we conducted an inter-rater reliability test during the training phase. Both nurses independently extracted data from a subset of the same medical records, and their results were compared. Agreement across key variables was evaluated using Cohen's kappa statistic, which demonstrated substantial agreement ($\kappa > 0.75$), indicating high inter-rater reliability.

Throughout the data collection period, close supervision and routine quality checks were conducted to ensure ongoing accuracy, completeness, and consistency of the data. Any discrepancies or uncertainties were promptly reviewed and resolved through discussion and feedback, further enhancing the validity and reliability of the dataset.

## Operational definitions

*Severe acute malnutrition*: Weight-for-height $< -3$ SD, mid-upper arm circumference of <11.5cm, or bilateral pitting oedema.

*Mortality:* Any death of a child under five years of age receiving treatment in the hospital.

*Time to death*: Time in days from the date of admission to the date of death.

*Censored*: Individuals whose time to death was not observed because of the termination of the study before the occurrence of death or because subjects left the study before death had occurred.

*Survival time*: The number of days that an individual child had survived in the hospital over the follow-up period from the date of admission for a disease until death or censorship had occurred.

## Variables

**Outcome variable.** The outcome variable was the "time to death" of an admitted child under five years of age.

**Event of interest.** Death of an admitted child under five years of age.

**Exposure variables.** Exposure variables included predictors of interest such as sex, child age, sex, breastfeeding status, referral status, HIV status, and administered treatments such as amoxicillin, oxygen therapy, and NG tube.

## Data management

We examined the raw data for errors and omissions. We checked the data and removed incorrect, redundant, and incomplete data. We cleaned up the data to maintain consensus. Besides this, we coded the data, entered it into the MS Excel spreadsheet, and then stored it on a laptop in a protected folder.

## Data analysis

Data from the MS Excel spreadsheet was directly imported into STATA 16 for analysis. Descriptive statistics such as frequencies, medians, and percentages were used to describe the data. We used Kaplan Meier (KM) curves and Log-rank tests to estimate survival probability and compare survival curves across distinct groups. Cox proportional hazard regression was used to identify predictors of mortality. Predictor variables with a p-value < 0.25 in the bivariable Cox proportional hazard regression model were entered into the multivariable Cox proportional hazard regression model to control for confounding. In the multivariable Cox proportional hazard model, we also included variables that demonstrated a significant association with the outcome of interest in previous studies. An adjusted hazard ratio with a 95% confidence interval presented the output of the analysis, and we declared it statistically significant when the p-value was less than 0.05.

## Ethical considerations

This study (protocol number: P.09/21/3415) was reviewed and approved by the Malawi's College of Medicine Research and Ethics Committee (COMREC) (certificate of ethical approval attached). Since we used secondary data, it was impractical to get individual patient consent. Therefore, we got an approval to use secondary data from the Mulanje District Research Coordinating Committee (approval letter attached). We also obtained an approval letter from the Kamuzu University of Health Sciences (approval letter attached). In this study, we used codes to identify subjects instead of collecting specific personal identifiers, thereby maintaining confidentiality. Since we used pre-existing data, there was no physical harm to the subjects.

## Results

### Participants

A total of 7,685 under-five children were admitted to Mulanje District Hospital between January 2017 and February 2021. After applying eligibility criteria, 6,257 records were retained. Of these, 1,041 children with surgical conditions or burns and 387 with incomplete data were excluded. Among the remaining 4,829 records, 643 were SAM cases and 5,614 were non-SAM.

Using simple random sampling, 227 SAM and 227 non-SAM children were selected for analysis. Each participant was followed up for 30 days from admission to determine survival status. Among SAM children, 19 (8.4%) died, while 208 survived. Among non-SAM children, 48 (21.2%) died, and 179 survived (Fig 1).

### Socio-demographic characteristics of the study participants

Table 1 presents the sociodemographic characteristics of the study population. The mean age distribution and admission patterns were similar between the groups. Male children were slightly more represented among non-SAM cases (60.8%) than among SAM cases (51.5%) (p = 0.047). Most admissions in both groups were primary admissions and self-referrals. Breastfeeding was more common among non-SAM children, while SAM children had a higher proportion with unknown vaccination status.

### Clinical characteristics of the study participants

Clinical characteristics differed significantly between groups (Table 2). Fever was more prevalent in non-SAM children (95.2%) than in SAM children (79.7%, p < 0.0001). Convulsions were markedly higher in non-SAM children (46.7%) than in SAM children (7.1%, p < 0.0001), likely reflecting the burden of severe malaria and other neurological conditions.

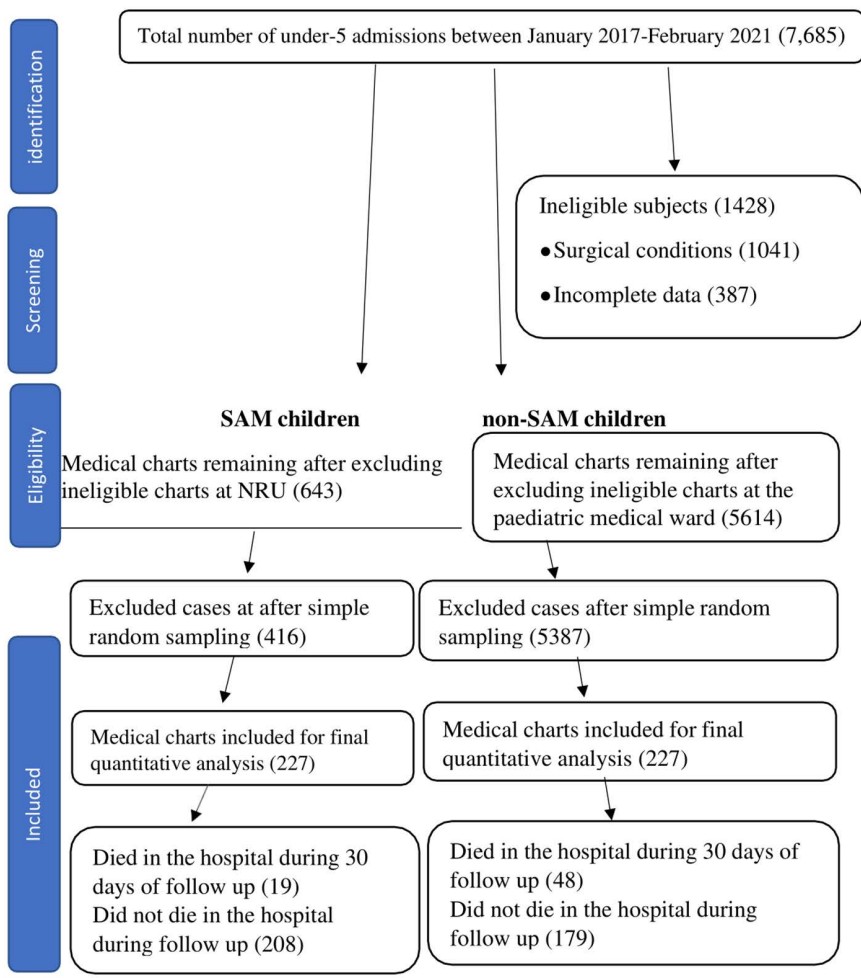

**Fig 1. Sampling flow for the selection of participants (Developed by authors).**

## Treatment administered

Treatment modalities varied significantly (Fig 2). Almost all SAM children (98.2%) received IV antibiotics, while 76.2% of non-SAM children did. Antimalarial treatment was more common among non-SAM children (67.8%) than SAM children (34.8%), aligning with the higher prevalence of malaria. Oral amoxicillin use was more frequent in SAM children (54.2%) compared to non-SAM children (34.8%).

## Follow-up duration

The median duration of hospital stay and follow-up was longer for SAM children (7 days) than for non-SAM children (2 days). The total cumulative follow-up time was 2,204 child-days, with 1,666 for SAM and 538 for non-SAM participants.

## Inpatient treatment outcomes

Treatment outcomes are summarized in Table 3. Mortality was significantly higher among non-SAM children (21.2%) compared to SAM children (8.4%) (p = 0.0003). Discharge rates were similar between the groups (~75%). However, SAM children had a significantly higher absconding rate (14.1% vs. 0.4%, p < 0.0001).

**Table 1. Sociodemographic characteristics between SAM and non-SAM children (n = 454).**

| Variable | Category | SAM children | Non-SAM children | Between group difference | |
|---|---|---|---|---|---|
| | | | | Mean or Proportion | p-value |
| Age | 0-23 months | 134(59.0%) | 128(56.4%) | −0.035 | 0.4490 |
| | 24-59 months | 93(41.0%) | 99(43.6%) | | |
| Sex | Male | 117(51.5%) | 138(60.8%) | 0.092 | 0.0470 |
| | Female | 110(48.5%) | 89(39.2%) | | |
| Referral status | Self-referral | 205(90.3%) | 200(88.1%) | −0.022 | 0.4495 |
| | Health-centre | 22(9.70%) | 27(11.9%) | | |
| Admission | Primary | 225(99.1%) | 224(98.7%) | −0.004 | 0.6529 |
| | Readmission | 2(0.90%) | 3(1.30%) | | |
| Breastfeeding | Yes | 105(46.3%) | 121(53.3%) | −0.004 | 0.9250 |
| | No | 122(53.7%) | 106(45.7%) | | |
| Vaccination | Complete | 64(28.2%) | 114(50.2%) | | |
| | Not completed | 56(28.6%) | 67(30.4%) | | |
| | Not received | 8(3.50%) | 4(1.80%) | | |
| | Unknown | 90(69.7%) | 40(17.2%) | | |

**Table 2. Comparison of clinical characteristics between SAM and non-SAM children (n = 454).**

| Variable | Category | SAM children | Non-SAM children | Between group difference | |
|---|---|---|---|---|---|
| | | | | Proportion | p-value |
| Fever | Yes | 181(79.7%) | 216(95.2%) | 0.154 | <0.0001 |
| | No | 46(20.3%) | 11(4.80%) | | |
| Vomiting | Yes | 90(39.7%) | 73(32.2%) | −0.074 | 0.0963 |
| | No | 137(60.3%) | 158(67.8%) | | |
| Diarrhoea | Yes | 169(74.5%) | 50(22.0%) | −0.524 | <0.0001 |
| | No | 58(25.5%) | 177(78.0%) | | |
| Anaemia | Yes | 82(36.1%) | 81(35.7%) | −0.004 | 0.9221 |
| | No | 145(63.9%) | 146(64.3%) | | |
| Cough | Yes | 89(39.2%) | 104(45.8%) | 0.066 | 0.1544 |
| | No | 138(60.8%) | 123(54.2%) | | |
| Convulsions | Yes | 16(7.10%) | 106(46.7%) | 0.396 | <0.0001 |
| | No | 221(92.9%) | 121(53.3%) | | |
| Oedema | Yes | 100(44.1%) | 3(1.30%) | −0.427 | <0.0001 |
| | No | 127(55.9%) | 224(98.7%) | | |

## Cause-specific mortality

Among the 454 children, the most common diagnoses were SAM (50%) and severe malaria (31.7%). Severe malaria accounted for the highest proportion of deaths (41.8%), followed by SAM (28.4%) (Table 4). Other causes included pneumonia, gastroenteritis, and sepsis.

## Mortality outcomes by nutrition status

Table 5 summarizes the mortality outcomes among under-five children by nutritional status. The overall inpatient mortality rate was 14.8% (67/454). Mortality was significantly higher in non-SAM children (21.1%) compared to SAM children

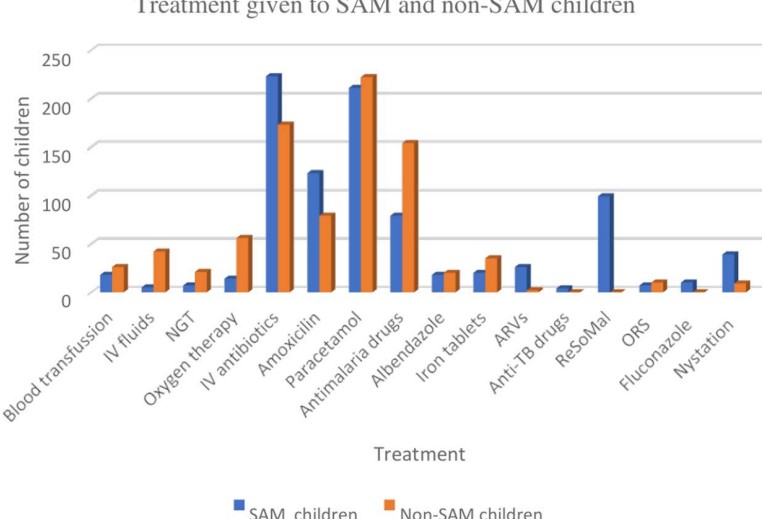

**Fig 2. Types of treatment given to SAM and non-SAM children (Developed by authors).**

**Table 3. Comparison of the treatment outcomes between SAM and non-SAM children (n = 454).**

| Outcome | SAM children | Non-SAM children | Between group difference | |
|---|---|---|---|---|
| | | | Proportion | p-value |
| **Died** | 19 (8.4%) | 48 (21.2%) | 0.118 | 0.0003 |
| **Recovered** | 171 (75.3%) | 172 (75.8%) | 0.004 | 0.9130 |
| **Absconded** | 32 (14.1%) | 1 (0.4%) | −0.136 | <0.0001 |
| **Referred** | 5 (2.2%) | 6 (2.6%) | 0.004 | 0.7602 |

(8.4%), and this difference was statistically significant (Z = 3.84, p < 0.001). The median time to death differed substantially between the two groups. Among SAM children, the median time to death was 5 days (IQR: 2–8), while among non-SAM children it was only 1 day (IQR: 1–2), suggesting that non-SAM children tended to present more acutely and deteriorated more rapidly.

### Kaplan-Meier survival estimates

The overall Kaplan-Meier (KM) survival estimate indicated that the probability of survival among under-five children remained high during the first 12 days of hospitalization but gradually declined with longer follow-up. The median survival time for the entire cohort was 22 days (Fig 3). When stratified by nutritional status, the KM survival curves showed a distinct pattern: SAM children consistently demonstrated higher survival probabilities than non-SAM children during the first 9 days of admission. This suggests that non-SAM children experienced earlier deterioration and death, while SAM children tended to survive longer in the initial phase of hospitalization (Fig 4).

### Predictors of mortality

Table 6 shows the results of Cox proportional hazards regression analyses. In SAM children, not receiving amoxicillin was a significant independent predictor of mortality (AHR: 4.59 (95% CI: 1.46–14.43), p = 0.009)). In non-SAM children, several factors independently predicted mortality, including facility referral (AHR 2.66 (95% CI: 1.34–5.27), p = 0.005)), oxygen

**Table 4. Cause-specific mortality (n = 454).**

| Variable | Survival status | | |
|---|---|---|---|
| | Censored | Died | Total |
| Acute bronchiolitis | 1(0.3%) | 1(1.5%) | 2(0.4%) |
| Acute gastroenteritis | 7(1.8%) | 6(9.0%) | 13(2.9%) |
| Diabetes Mellitus (DM)I | 0(0.00) | 1(1.5%) | 1(0.2%) |
| Effective endocarditis | 1(0.3%) | 0(0.00%) | 1(0.4%) |
| Meningitis | 1(0.3%) | 1(1.5%) | 2(0.4%) |
| SAM | 208(53.8%) | 19(28.4%) | 227(50%) |
| Sepsis | 20(5.2%) | 2(3.0%) | 22(4.9%) |
| Severe asthma attack | 2(0.5%) | 0(0.00%) | 2(0.4%) |
| Severe pneumonia | 30(7.7%) | 8(11.9%) | 38(8.37%) |
| Severe malaria | 116(30.0%) | 28(41.8%) | 144(31.7%) |
| Ventricular Septal Defect | 0(0.00%) | 1(1.5%) | 1(0.2%) |
| Typhoid | 1(0.26%) | 0(0.00%) | 1(0.2%) |

**Table 5. Mortality outcomes by nutritional status (n = 454).**

| Variable | Result | Statistical context |
|---|---|---|
| **Mortality rate (%)** | Non-SAM: 21.1% (48/227)<br>SAM: 8.4% (19/227)<br>Total: 14.8% (67/454) | $Z = 3.84$, $p < 0.001$ |
| **Hazard ratio (SAM vs non-SAM)** | 0.15 (95% CI: 0.08–0.28) | $\chi^2 = 14.73$, $p < 0.0001$ |
| **Median time to death** | SAM: 5 days (IQR: 2–8)<br>Non-SAM: 1 day (IQR: 1–2) | |

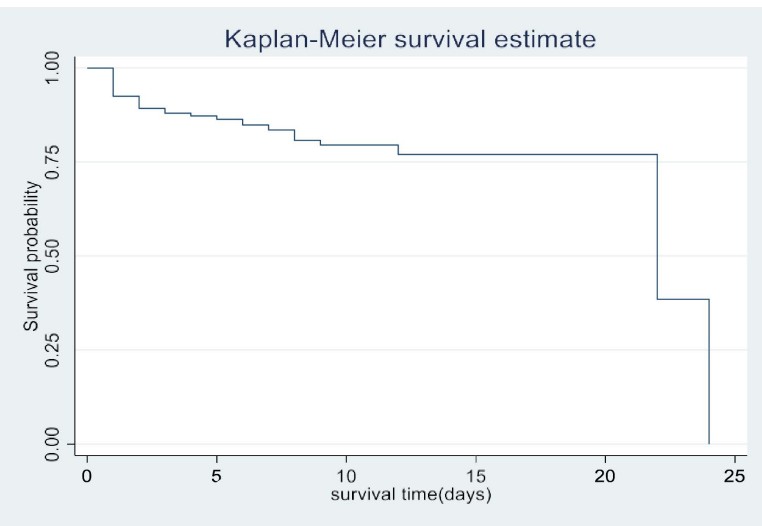

**Fig 3. An overall Kaplan-Meier survival graph for all in-patient paediatric admissions at Mulanje District Hospital.**

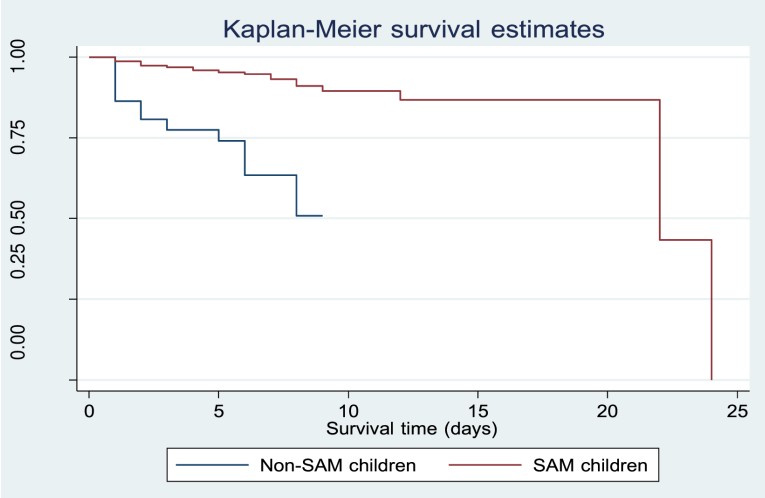

**Fig 4. The Kaplan-Meier survival graphs for SAM and non-SAM children at Mulanje District Hospital.**

**Table 6. Predictors of mortality among SAM and non-SAM children (n = 454).**

| Variable | Category | SAM children | | | | Non-SAM children | | | |
|---|---|---|---|---|---|---|---|---|---|
| | | CHR (95%CI) | P value | AHR (95%CI) | P value | CHR (95%CI) | P value | AHR (9%5 CI) | P value |
| **Sex** | Male<br>Female | 1.20 (0.47-3.04)<br>1 | 0.699 | 1.52 (0.56-4.13)<br>1 | 0.408 | 1.05 (0.58-1.88)<br>1 | 0.859 | 0.91 (0.49-1.69)<br>1 | 0.784 |
| **Age** | 0-23 months<br>24-59 months | 0.63 (0.24-1.64)<br>1 | 0.350 | 0.99 (0.25-3.88)<br>1 | 0.995 | 1.16 (0.65-2.07)<br>1 | 0.604 | 0.58 (0.12-2.80)<br>1 | 0.449 |
| **Breastfeeding status** | Breastfed<br>Not breastfed | 0.52 (0.20-1.35)<br>1 | 0.181 | 0.65 (0.17-2.41)<br>1 | 0.526 | 1.41 (0.78-2.53)<br>1 | 0.249 | 2.23 (0.46-10.83)<br>1 | 0.316 |
| **Referral status** | Facility referral<br>Self –referral | 1.25 (0.28-5.48)<br>1 | 0.765 | 1.77 (0.38-8.19)<br>1 | 0.464 | 3.08 (1.58-6.00)[*]<br>1 | 0.001 | 2.66 (1.34-5.27)[*]<br>1 | 0.005 |
| **HIV status** | Positive<br>Negative | 1.32 (0.38-4.63)<br>1 | 0.655 | 1.40 (0.37-5.35)<br>1 | 0.615 | 1.46 (0.20-10.65)<br>1 | 0.706 | 1.07 (0.13-8.65)<br>1 | 0.321 |
| **Amoxicillin** | Not receiving<br>Receiving | 4.69 (1.54-14.29)*<br>1 | 0.006 | 4.59 (1.46-14.43)*<br>1 | 0.009 | 30.32 (4.17-220.2)<br>1 | 0.001 | 33.49 (4.47-45.07)*<br>1 | 0.001 |
| **NGT** | Inserted<br>Not inserted | 4.34 (0.98-19.11)<br>1 | 0.052 | 3.30 (0.51-21.13)<br>1 | 0.207 | 3.90 (2.05-7.42)*<br>1 | <0.0001 | 1.28 (0.61-2.67)<br>1 | 0.499 |
| **Oxygen therapy** | Receiving<br>Not receiving | 2.89 (0.75-11.08)<br>1 | 0.120 | 2.03 (0.42-9.75)<br>1 | 0.372 | 4.12 (2.31-7.33)*<br>1 | <0.0001 | 4.04 (2.11-7.71)*<br>1 | <0.0001 |

[2] Abbreviations: Crude Hazard Ratio (CHR), Confidence Interval (CI), Adjusted Hazard Ratio (AHR). Cox proportional hazard regression at bi-variable and multivariable level.

1 = Reference category. Note: * p-value < 0.05.

therapy (AHR 4.04 (95% CI: 2.11–7.71), p < 0.0001)) and not receiving amoxicillin: AHR 33.49 (95% CI: 4.47–250.7), p = 0.001.

The extremely large effect size associated with not receiving amoxicillin in non-SAM children suggests strong clinical significance. This finding indicates that early antibiotic administration may be life-saving in this group, possibly due to the high burden of severe bacterial infections such as pneumonia and sepsis, which progress rapidly without prompt treatment. The strength of this association, even after adjusting for confounders, underscores the critical role of timely empirical antibiotic therapy in reducing mortality risk.

## Discussion

In this study, the inpatient mortality rate for all under-five admissions was 14.8%. This rate is disproportionately high compared to those observed in studies done in Iran and South Sudan, whose mortality rates were 1.35% and 5.7%, respectively [28,29]. The higher incidence of mortality in the present study could be attributed to several factors, including inadequate case management, late presentation, delayed initiation of care and treatment, inadequate staffing, and a lack of essential medical supplies.

Contrary to the WHO's report [6], the majority of deaths in the present study occurred among non-SAM children rather than those with SAM. This unexpected trend may be attributed to the acute and rapidly fatal nature of illnesses such as severe malaria, pneumonia, or sepsis, which were more prevalent among non-SAM children. The shorter median time to death among this group further suggests that many children arrived at the hospital in critical condition, with limited opportunity for effective intervention.

Another contributing factor could be delays and systemic challenges in the referral process. Among non-SAM children, facility referral emerged as a significant predictor of mortality. This likely reflects issues such as late presentation, poor transport infrastructure, lack of pre-referral stabilization, and inadequate triage at lower-level facilities. Many children may have deteriorated during transit or arrived too late for life-saving treatment, underscoring the urgent need for strengthening early recognition, referral pathways, and emergency care capacity.

Our study revealed that children with SAM had a longer median time to death (5 days) compared to non-SAM children (1 day). However, this disparity in time to death does not imply that SAM is less critical. Rather, the difference likely reflects variations in underlying biological mechanisms, distinct causes of mortality, and differences in medical interventions. The slower progression to death in SAM children may be due to physiological adaptations to prolonged malnutrition. In contrast, non-SAM children often succumb more quickly to acute, rapidly fatal conditions such as sepsis or malaria, which can cause sudden systemic failure. Additionally, SAM children typically receive intensive medical care, including nutritional therapy, routine antibiotics, and treatment of complications, which may extend survival [30]. Non-SAM children, however, might not receive such interventions or may access care too late, leading to a shorter time to death.

This study found that a higher proportion of non-SAM children, many of whom had severe malaria, died compared to SAM children. In sub-Saharan Africa, severe malaria is a recognized leading cause of pediatric mortality, and it might have contributed to the observed patterns of mortality [31]. Despite hospital treatment, severe malaria's acute complications, such as cerebral involvement, severe anemia, and respiratory distress, can cause rapid clinical deterioration and death. The higher in-hospital mortality seen in the non-SAM group may have been largely caused by these factors. In contrast, while SAM children were critically ill, they often received targeted therapeutic feeding and supportive care, which might have buffered early mortality during admission.

The present study found that the discharge rates were similar between the SAM and non-SAM groups, indicating that they recovered similarly while still in the hospital. Notably, the study did not track post-discharge outcomes for either group, limiting the ability to assess how post-discharge factors may have influenced the observed mortality difference. Existing research, however, indicates that the post-discharge mortality rate is 2-fold higher in SAM children as compared to non-SAM children [32]. This is probably because of persistent immunosuppression, ongoing nutritional deficiencies, and

restricted access to continuity of care. Without accounting for these delayed risks, the lower in-hospital mortality observed among SAM children might underestimate their true vulnerability over time.

Sex has been identified as a potential predictor of mortality in children with SAM, with a study in Ethiopia showing higher mortality risk in male children [33]. This is aligned with the fact that males are biologically weaker than females, making them more susceptible to life risks, including death [34]. However, the present analysis found no significant association between sex and mortality in SAM and non-SAM children. The possible cause for this variation could be differences in sample size, study setting and the severity of the disease.

Consistent with another study [35], our research found that age was not a predictor of mortality in SAM and non-SAM children. This contrasts with studies in Egypt [36] and Ethiopia [37], where age was a significant predictor. In Ethiopia [37], children under 24 months were 2.84 times more likely to die than those over 24 months. Literature has shown that infants possess weak and underdeveloped immune systems, making them more susceptible to severe diseases and death [38,39].

Unlike in SAM children, facility referral was a significant predictor of death among non-SAM children. Non-SAM children who were referred from health facilities had a 2.6-fold increased risk of death as compared to the self-referrals. This could be because patients from health facilities were reporting late while the condition was very critical and had a poor prognosis. This is similar to a study done in Congo, where facility referrals were significantly associated with late referrals and subsequent child deaths [40].

It is scientifically known that antibiotics, including amoxicillin, reduce mortality in children with SAM [21]. This study observed that both SAM and non-SAM children who did not receive amoxicillin had an increased risk of death. This is similar to an Ethiopian study where having not received antibiotics, including amoxicillin was also a significant predictor of death in under-five children [41]. The possible reason to explain this could be the fact that amoxicillin, just like any other antibiotic, improves the clinical response to life-threatening infections, therefore reducing the risk of mortality [42,43].

However, the growing threat of antimicrobial resistance (AMR) must be weighed against the expanding empirical use of amoxicillin. Due in part to widespread empirical prescribing and subpar drug quality, regional surveillance indicates that common pediatric pathogens are becoming less susceptible to penicillins [44]. Therefore, to maintain the drug's proven survival benefit and long-term effectiveness, routine amoxicillin administration should be combined with antimicrobial stewardship safeguards, such as strict clinical criteria for initiation, procurement of quality-assured formulations, and regular updates to local treatment protocols based on AMR data.

Contrary to the results observed among SAM children, having received oxygen therapy was a predictor of mortality among non-SAM children. Our study showed that non-SAM children who received oxygen therapy were 4 times more likely to die than those patients who had not received oxygen therapy. The association suggests that poor outcomes may partly result from oxygen toxicity, although it could also reflect confounding by severity, as oxygen is typically given to critically ill patients. Excessive or prolonged oxygen use can cause hyperoxia, leading to oxidative stress, inflammation, and organ damage. In young children, this may result in lung injury, retinopathy, or multi-organ failure, particularly when high-flow oxygen is used without proper monitoring. [45,46]. A similar relationship was found in a study done in the United States of America, where oxygen therapy was associated with increased inpatient mortality in children because of hyperoxia [47]. Our study suggested that nurses and clinicians should use a cautious, evidence-based approach to oxygen therapy, aided by pulse oximetry whenever feasible, to reduce the risk of oxygen toxicity. Oxygen should be initiated only when clear indications exist (e.g., $SpO_2 < 90\%$ or signs of respiratory distress) and should be titrated to maintain saturations within the recommended target range (90–95%).

NGT insertion is a key factor associated with mortality in hospitalized children, with literature linking improper insertion to complications such as tracheal intubation, pneumothorax, ventilatory failure, and death [48]. A report has shown that NGT insertion is mostly used in critically ill children with poor prognosis and hence higher risk of death [49]. A study in Ethiopia found NGT insertion to be a significant predictor of mortality in SAM children [50]. However, this was not the case

in the present study, where NGT insertion did not significantly predict mortality in either SAM or non-SAM children. This discrepancy may be due to differences in study setting, caseload, sample size, and disease severity.

The association between HIV infection and mortality in SAM has been described in several studies [40,51]. Research has aligned this association with nutritional acquired immune deficiency syndrome (NAIDS) which worsens the fragility of a deteriorating immune system [52]. A study has shown that HIV-infected children with SAM are three times more likely to die as compared to HIV-negative children [51]. However, the present study showed that HIV was not a significant predictor of mortality for SAM and non-SAM children. This study did not investigate why such was the case.

## Strengths and limitations

In this study, we applied the right sampling technique, the right sample size, and accurate research procedures. We used a simple random sampling technique, therefore preventing selection bias. Besides this, our sample size was large and representative of the target population, hence strengthening the power of the study. However, we acknowledged the presence of a few limitations. Our study was affected by unmeasured confounding from the unmeasured variables. In this study, we did not investigate the predictive effects of biochemical and health system factors. Additionally, the high mortality rate observed in non-SAM children may have been influenced by the confounding effects of severe malaria, potentially leading to an overestimation of the mortality rate in non-SAM children compared to SAM children. This study excluded surgical admissions, which may have introduced selection bias. Surgical conditions often carry high mortality risk, and their exclusion may have influenced the observed mortality rates in non-SAM children by disproportionately including acutely ill medical cases. As such, caution is needed when generalizing these findings to the broader under-five inpatient population. Lastly, the study did not track post-discharge outcomes for either group, limiting the ability to assess how post-discharge factors may have influenced the observed mortality difference

## Conclusion and recommendations

This study showed a longer median time to death among SAM children compared to non- SAM children. Among SAM children, not receiving amoxicillin was a significant predictor of mortality. Among non-SAM children, facility referral, receiving oxygen therapy, and not receiving amoxicillin were found to be significant predictors of mortality. Therefore, to reduce under-five deaths, clinical and nursing interventions must target these modifiable risk factors and ensure that the health-system inputs required to implement those interventions are reliably available

The present results also bring up significant issues regarding the antibiotics stewardship. More research is necessary to weigh the risk of antibiotic resistance against life-saving empirical therapy, given the observed association between amoxicillin and survival. National antimicrobial stewardship strategies should be implemented in tandem with efforts to guarantee the proper access and prudent use of antibiotics such as amoxicillin. Furthermore, the association between oxygen treatment and death emphasizes the necessity of protocol changes and increased clinical supervision. In particular, oxygen should only be used when hypoxia is confirmed or highly suspected, and should be titrated according to clinical severity and pulse oximetry. To prevent both underuse and overuse, it is critically necessary to train staff and standardize oxygen initiation criteria, monitoring, and weaning procedures

We also suggested that an analytical study that includes multidimensional predictors of mortality is needed to prevent confounding effects originating from unmeasured confounders. Additionally, the study recommends that future research should track post-discharge outcomes for both groups in order to evaluate the impact of post-discharge factors on the mortality differences between the groups.

## Author contributions

**Conceptualization:** Masuzgo Rodney Pajogo, Miriam Ndholvu.

**Data curation:** Masuzgo Rodney Pajogo.

**Formal analysis:** Masuzgo Rodney Pajogo.

**Funding acquisition:** Masuzgo Rodney Pajogo.

**Investigation:** Masuzgo Rodney Pajogo.

**Methodology:** Masuzgo Rodney Pajogo.

**Project administration:** Masuzgo Rodney Pajogo.

**Resources:** Masuzgo Rodney Pajogo.

**Software:** Masuzgo Rodney Pajogo.

**Supervision:** Masuzgo Rodney Pajogo.

**Validation:** Masuzgo Rodney Pajogo.

**Visualization:** Masuzgo Rodney Pajogo.

**Writing – original draft:** Masuzgo Rodney Pajogo.

**Writing – review & editing:** Masuzgo Rodney Pajogo, Miriam Ndholvu.

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
