## [Decision Letter · Decision Letter 0]

16 Jun 2024

PONE-D-24-00702Time to death and predictors of mortality among under-five children with severe acute malnutrition hospitalized at Mulanje District Hospital in Southern Malawi; A retrospective cohort study.PLOS ONE

Dear Dr. Pajogo,

Thank you for submitting your manuscript to PLOS ONE. After careful consideration, we feel that it has merit but does not fully meet PLOS ONE’s publication criteria as it currently stands. Therefore, we invite you to submit a revised version of the manuscript that addresses the points raised during the review process.

Please clarify the study objective (s) and ensure it aligns with the study design, the analytic approach and the way the results are presented.

We look forward to receiving your revised manuscript.

Kind regards,

Calistus Wilunda, DrPH

Academic Editor

PLOS ONE

Journal Requirements:

2. In the online submission form, you indicated that "Data will be made available upon communicating with the corresponding author."

4.  If any Table files show as item type ‘figure/other’ please change item type to ‘Table’.

Additional Editor Comments:

Please clarify the study objective and ensure the design and analysis align with the study objective. The title and the objective in the abstract suggest that the study population is children with SAM. It is unclear why children without SAM were included in this study. The mention of the exposed and un-exposed groups based on SAM status is confusing. The exposures should be the predictors of interest. In the regression model, consider that some variables may be on the causal pathway and including then in the same model may attenuate the HRs of distal variables.

Reviewers' comments:

Reviewer's Responses to Questions

**Comments to the Author**

1. Is the manuscript technically sound, and do the data support the conclusions?

Reviewer #1: No

Reviewer #2: Yes

2. Has the statistical analysis been performed appropriately and rigorously? 

Reviewer #1: I Don't Know

Reviewer #2: Yes

3. Have the authors made all data underlying the findings in their manuscript fully available?

Reviewer #1: Yes

Reviewer #2: No

4. Is the manuscript presented in an intelligible fashion and written in standard English?

Reviewer #1: Yes

Reviewer #2: Yes

5. Review Comments to the Author

Reviewer #1: Thank you for asking me to review this manuscript on a very important subject; Time to death and predictors of mortality among under five children admitted with severe acute malnutrition.

The Abstract introduces the subject very well and clearly describes what was done overall. The conclusions going by the introduction in the abstract also seem appropriate.

I however found a number of issues with the Manuscript that the Authors need to address.

In the Introduction, it does not come out clearly what the Authors set out to do in their study. Their inclusion and exclusion criteria are also not clear to the reader. They talk about patients who were excluded but don't state the age of these patients. They even talk about 'incomplete charts' being excluded. This section seriously needs revision. One would expect that in such a study that was looking at children with SAM, those without SAM would be excluded, which wasn't the case.

On Sampling, it is not clear to me how the sampling was done once the Authors came out with a SAM group and a non SAM group. In any case, it doesn't come out clearly in their title and introduction, that their study is about comparing predictors of mortality in children with SAM and those without SAM. I thought that going by the title of their work, children without SAM should have automatically been excluded. This makes all their subsequent writing about SAM and non SAM disconnected from the title of their work.

In Figure 3, Figure 4 and Table 5, the Authors now introduce a new concept, Survival function that wasn't any where in their 'not so clear' objectives. This is misplaced.

In the results section, The Authors bring out the Mortality rate and time to death. They also bring about the predictors of mortality but present them as if they had set out to compare predictors of mortality in children with SAM and in children without SAM. if this was the intention, it should be clarified right from the title of the study to the introduction and methods.

When they get to discuss their results, the Authors discuss the predictors of mortality from a point of why they are different between the two populations. I think they should be discussing why shock and not using Amoxicillin were predictive of mortality and also relating to other studies

The Authors also dedicate three Paragraphs in the discussion section to discussing what was not predictive of mortality in their study before finally discussing what was. This needs serious revision. The entire discussion section seems disconnected from the objectives of the study, if I got them well from the Abstract.

Additionally, when discussing about shock, they focus on discussing shock in non SAM children not children with SAM who are the major focus of the study.

The conclusion then becomes flawed because it is seemingly based on the finding of shock in children without SAM. This effectively disconnects the conclusion from the title. Besides, the Authors pool all predictors of mortality (in SAM and non SAM) children together, which creates the impression that all the variables given are predictors of mortality in children with SAM which is not true.

In conclusion, the Authors need to decide what their work was about and be clear about what they set out to do, how they did it and what their conclusions were.

Reviewer #2: It is an insightful paper, especially because of its clarity and replicability. The methodology is well-articulated, facilitating easy comprehension and replication by other researchers. Moreover, the methodology and findings are robust.

However, there is room for further elaboration on discussion on the disparities between SAM (Severe Acute Malnutrition) and non-SAM groups concerning the contrasting results between the two groups of samples.

1. The paper could report descriptive statistics to compare if the average or median values of predictors and outcomes between the two groups are statistically different. This analysis would explain any significant differences between SAM and non-SAM cohorts. strengthen its statistical validity.

2. It needs further elaboration on SAM vs. non-SAM disparities in terms of their treatments. Understanding these distinctions is crucial for comprehending the observed outcomes. Highlighting the specific interventions, nutritional strategies, and medical protocols tailored for each group would enhance the paper's comprehensiveness.

3. It's essential to address why this research is significant in the context of existing literature and how its findings can inform healthcare policies and clinical practices. Discussing potential interventions or guidelines stemming from the study's results would augment its impact.

6. PLOS authors have the option to publish the peer review history of their article (what does this mean? ). If published, this will include your full peer review and any attached files.

**Do you want your identity to be public for this peer review?** For information about this choice, including consent withdrawal, please see our Privacy Policy .

Reviewer #1: No

Reviewer #2: No

---

## [Author Response · Author response to Decision Letter 1]

11 Aug 2024

Response to reviewers' and editors' comments has been attached

---

## [Decision Letter · Decision Letter 1]

4 Mar 2025

PONE-D-24-00702R1Comparison of time to death and predictors of mortality between under-five children suffering from severe acute malnutrition and under-five children without severe acute malnutrition hospitalized at Mulanje District Hospital in Southern MalawiPLOS ONE

Dear Dr. Pajogo,

Thank you for submitting your manuscript to PLOS ONE. After careful consideration, we feel that it has merit but does not fully meet PLOS ONE’s publication criteria as it currently stands. Therefore, we invite you to submit a revised version of the manuscript that addresses the points raised during the review process. Please submit your revised manuscript by Apr 13 2025 11:59PM. If you will need more time than this to complete your revisions, please reply to this message or contact the journal office at plosone@plos.org . Please include the following items when submitting your revised manuscript:

We look forward to receiving your revised manuscript.

Kind regards,

Helen Howard

Staff Editor

PLOS ONE

**Additional Editor Comments:**

The manuscript has been evaluated by one reviewer, and their comments are available below.

The reviewers have raised a number of concerns that need attention. They request clarification on the methods, and further discussion in the manuscript.

Could you please revise the manuscript to carefully address the concerns raised?

Reviewers' comments:

Reviewer's Responses to Questions

**Comments to the Author**

1. If the authors have adequately addressed your comments raised in a previous round of review and you feel that this manuscript is now acceptable for publication, you may indicate that here to bypass the “Comments to the Author” section, enter your conflict of interest statement in the “Confidential to Editor” section, and submit your "Accept" recommendation.

Reviewer #2: All comments have been addressed

2. Is the manuscript technically sound, and do the data support the conclusions?

Reviewer #2: Partly

3. Has the statistical analysis been performed appropriately and rigorously? 

Reviewer #2: N/A

4. Have the authors made all data underlying the findings in their manuscript fully available?

Reviewer #2: Yes

5. Is the manuscript presented in an intelligible fashion and written in standard English?

Reviewer #2: Yes

6. Review Comments to the Author

Reviewer #2: The paper titled "Comparison of Time to Death and Predictors of Mortality Between Under-Five Children Suffering from Severe Acute Malnutrition and Under-Five Children Without Severe Acute Malnutrition Hospitalized at Mulanje District Hospital in Southern Malawi" has been addressing several issues. However, several areas require improvement for clarity and completeness:

Title: The current title is quite lengthy and specific. A more concise and focused title would enhance readability and better capture the essence of the study.

Follow-Up Time: The term "total follow-up time" needs clarification. It is important to specify whether this refers to the cumulative number of days across all children in the study or the duration each individual child was monitored.

Inpatient Treatment Outcome: The statement regarding "one-third" is ambiguous. Is it suppose to be two-third?

Discharge Proportions: Although the discharge proportions are similar between the two groups (SAM and non-SAM), the mortality rates differ significantly. It would be helpful to explore whether the outcomes for non-SAM children post-discharge were tracked and if these might influence the observed mortality differences.

Malaria as a Confounder: The higher mortality rate observed in non-SAM children may be influenced by severe malaria, which introduces a potential confounding factor. This should be explicitly acknowledged as a limitation of the study, with a discussion on how it impacts the comparability between groups.

Non-Significant Variables: The analysis primarily focuses on significant variables while excluding non-significant ones. It would be beneficial to discuss why certain variables, particularly those related to malaria, were not significant and how this might affect the overall findings.

Discussion: The discussion section could benefit from a more comprehensive exploration of the results.

7. PLOS authors have the option to publish the peer review history of their article (what does this mean? ). If published, this will include your full peer review and any attached files.

**Do you want your identity to be public for this peer review?** For information about this choice, including consent withdrawal, please see our Privacy Policy .

Reviewer #2: No

---

## [Author Response · Author response to Decision Letter 2]

11 Mar 2025

Response to reviewer's comments have been attached

---

## [Decision Letter · Decision Letter 2]

10 Apr 2025

PONE-D-24-00702R2Comparison of time to death and its predictors in hospitalized children with and without severe acute malnutrition at Mulanje District Hospital, Southern MalawiPLOS ONE

Dear Dr. Pajogo,

Thank you for submitting your manuscript to PLOS ONE. After careful consideration, we feel that it has merit but does not fully meet PLOS ONE’s publication criteria as it currently stands. Therefore, we invite you to submit a revised version of the manuscript that addresses the points raised during the review process.

We look forward to receiving your revised manuscript.

Kind regards,

Abera Lambebo Temamo, Ph.D.

Academic Editor

PLOS ONE

Journal Requirements:

Reviewers' comments:

Reviewer's Responses to Questions

**Comments to the Author**

1. If the authors have adequately addressed your comments raised in a previous round of review and you feel that this manuscript is now acceptable for publication, you may indicate that here to bypass the “Comments to the Author” section, enter your conflict of interest statement in the “Confidential to Editor” section, and submit your "Accept" recommendation.

Reviewer #2: All comments have been addressed

2. Is the manuscript technically sound, and do the data support the conclusions?

Reviewer #2: Yes

3. Has the statistical analysis been performed appropriately and rigorously? 

Reviewer #2: Yes

4. Have the authors made all data underlying the findings in their manuscript fully available?

Reviewer #2: Yes

5. Is the manuscript presented in an intelligible fashion and written in standard English?

Reviewer #2: Yes

6. Review Comments to the Author

Reviewer #2: Overall, the authors have made substantial improvements to the manuscript, enhancing its clarity and rigor. However, it is necessary to address the potential for misinterpretation regarding the observed difference in time to death between children with Severe Acute Malnutrition (SAM) (5 days) and Non-SAM (1 day). To prevent the misconception that SAM is "less severe" or "less critical" than Non-SAM, the authors should explicitly emphasize that this disparity in time to death does not indicate a lower severity of SAM. Instead, it likely reflects differences in underlying biological mechanisms, distinct etiologies of mortality, and variations in medical interventions. Author may elaborate para 1-2 in the discussion section based on the finding such as the differing impacts of SAM and Non-SAM, where the process of death in children with SAM may be slower due to their bodies having adapted to severe malnutrition. In contrast, Non-SAM children may die more quickly due to highly fatal acute illnesses (such as sepsis and malaria) that cause rapid systemic damage and shock. Children with SAM also receive intensive medical care (such as nutritional therapy, antibiotics, or other supportive treatments), which may prolong their survival time. On the other hand, Non-SAM children may not receive the same interventions or may receive treatment too late, resulting in a faster time to death.

By incorporating these points, the authors can provide a more nuanced interpretation of their findings, ensuring that readers do not erroneously infer that SAM is less severe than Non-SAM based solely on the observed differences in time to death.

7. PLOS authors have the option to publish the peer review history of their article (what does this mean? ). If published, this will include your full peer review and any attached files.

**Do you want your identity to be public for this peer review?** For information about this choice, including consent withdrawal, please see our Privacy Policy .

Reviewer #2: No

---

## [Author Response · Author response to Decision Letter 3]

10 Apr 2025

A document containing responses to reviewers' comments has been attached

---

## [Decision Letter · Decision Letter 3]

30 Jun 2025

PONE-D-24-00702R3Comparison of time to death and its predictors in hospitalized children with and without severe acute malnutrition at Mulanje District Hospital, Southern MalawiPLOS ONE

Dear Dr. Pajogo,

Thank you for submitting your manuscript to PLOS ONE. After careful consideration, we feel that it has merit but does not fully meet PLOS ONE’s publication criteria as it currently stands. Therefore, we invite you to submit a revised version of the manuscript that addresses the points raised during the review process.

We look forward to receiving your revised manuscript.

Kind regards,

Guy Franck Biaou ALE, PhD

Academic Editor

PLOS ONE

Reviewers' comments:

Reviewer's Responses to Questions

**Comments to the Author**

1. If the authors have adequately addressed your comments raised in a previous round of review and you feel that this manuscript is now acceptable for publication, you may indicate that here to bypass the “Comments to the Author” section, enter your conflict of interest statement in the “Confidential to Editor” section, and submit your "Accept" recommendation.

Reviewer #2: All comments have been addressed

Reviewer #3: (No Response)

2. Is the manuscript technically sound, and do the data support the conclusions?

Reviewer #2: Yes

Reviewer #3: (No Response)

3. Has the statistical analysis been performed appropriately and rigorously? 

Reviewer #2: Yes

Reviewer #3: No

4. Have the authors made all data underlying the findings in their manuscript fully available?

Reviewer #2: Yes

Reviewer #3: Yes

5. Is the manuscript presented in an intelligible fashion and written in standard English?

Reviewer #2: Yes

Reviewer #3: No

6. Review Comments to the Author

Reviewer #2: (No Response)

Reviewer #3: Comments to the authors

In this study, the authors tried to address a critical gap in pediatric care in resource-limited settings. Its robust methodology and focus on understudied predictors are strengths.

The topic has the potential of contributing to the existing scientific knowledge on time to death among children with SAM and non-SAM.

I have the following major concerns, which the authors need to consider substantially.

Abstract

1. Overly dense; could benefit from brevity. The higher mortality in non-SAM children is not contextualized, which might confuse readers given SAM’s established risks.

2. Clarify why non-SAM mortality exceeds SAM despite SAM’s known severity.

Introduction

3. Transition from global to local context is abrupt. Limited data on Malawi’s SAM mortality post-2015.

4. Add recent Malawi-specific mortality statistics. Emphasize the clinical relevance of comparing SAM/non-SAM predictors.

Methods

5. Missing data handling and potential biases (e.g., selection bias due to excluded surgical cases) underdiscussed.

6. No mention of inter-rater reliability for data collectors.

7. Address how missing data were managed.

8. Justify exclusion of surgical patients more thoroughly.

9. Describe training for data collectors to ensure consistency.

Results

10. Generally the result presentation is not organized and needs major revision.

11. Extremely wide CI for non-SAM amoxicillin predictor (33.49; 4.47–250.7) suggests imprecision.

12. Mortality rate discrepancies (non-SAM > SAM) are not statistically contextualized.

13. Include visual aids (e.g., survival curves). Consider also description of the non-parametric use of the survival curves in the method section.

14. Discuss the clinical significance of large effect sizes (e.g., amoxicillin’s role in non-SAM).

Discussion

15. Under addresses why non-SAM mortality surpasses SAM despite SAM’s severity. Limited exploration of oxygen therapy’s risks (e.g., toxicity).

16. Expand on confounding factors (e.g., severe malaria’s role in non-SAM deaths). Discuss antibiotic resistance implications for amoxicillin recommendations.

Conclusions and Recommendations

17. Overlooks antibiotic stewardship concerns.

18. Recommendations lack specificity (e.g., how to optimize oxygen therapy).

19. Advocate for protocol revisions (e.g., timing of antibiotics, oxygen titration).

20. Highlight need for resource allocation (e.g., amoxicillin access).

7. PLOS authors have the option to publish the peer review history of their article (what does this mean? ). If published, this will include your full peer review and any attached files.

Reviewer #2: No

Reviewer #3: Yes: Dr. Dereje Tsegaye

---

## [Decision Letter · Decision Letter 4]

9 Sep 2025

PONE-D-24-00702R4Comparison of time to death and its predictors in hospitalized children with and without severe acute malnutrition at Mulanje District Hospital, Southern MalawiPLOS ONE

Dear Dr. Pajogo,

Thank you for submitting your manuscript to PLOS ONE. After careful consideration, we feel that it has merit but does not fully meet PLOS ONE’s publication criteria as it currently stands. Therefore, we invite you to submit a revised version of the manuscript that addresses the points raised during the review process.

We look forward to receiving your revised manuscript.

Kind regards,

Jenna Scaramanga

Staff Editor

PLOS ONE

Journal Requirements:

Additional Editor Comments:

Your manuscript is almost ready for acceptance. However, we found one statement which we feel is unsupported: "This is aligned to the fact that males are biologically weaker than females, making them more susceptible to life risks, including death [28]". This is not supported by the cited study (nor would the reverse claim be supported by it). Consider removing this statement, or revising it to reflect better the findings of the referenced article. 

Reviewer's Responses to Questions

**Comments to the Author**

1. If the authors have adequately addressed your comments raised in a previous round of review and you feel that this manuscript is now acceptable for publication, you may indicate that here to bypass the “Comments to the Author” section, enter your conflict of interest statement in the “Confidential to Editor” section, and submit your "Accept" recommendation.

Reviewer #2: (No Response)

2. Is the manuscript technically sound, and do the data support the conclusions?

Reviewer #2: (No Response)

3. Has the statistical analysis been performed appropriately and rigorously? 

Reviewer #2: (No Response)

4. Have the authors made all data underlying the findings in their manuscript fully available?

Reviewer #2: (No Response)

5. Is the manuscript presented in an intelligible fashion and written in standard English?

Reviewer #2: (No Response)

6. Review Comments to the Author

Reviewer #2: (No Response)

7. PLOS authors have the option to publish the peer review history of their article (what does this mean? ). If published, this will include your full peer review and any attached files.

**Do you want your identity to be public for this peer review?** For information about this choice, including consent withdrawal, please see our Privacy Policy .

Reviewer #2: No

---

## [Author Response · Author response to Decision Letter 5]

9 Sep 2025

Response letter has been attached

---

## [Editor Report · Decision Letter 5]

16 Sep 2025

Comparison of time to death and its predictors in hospitalized children with and without severe acute malnutrition at Mulanje District Hospital, Southern Malawi

PONE-D-24-00702R5

Dear Dr. Pajogo,

We’re pleased to inform you that your manuscript has been judged scientifically suitable for publication and will be formally accepted for publication once it meets all outstanding technical requirements.

Kind regards,

Laura Kelly, PhD

Division Editor

PLOS One
---

## [Editor Report · Acceptance letter]

PONE-D-24-00702R5

PLOS ONE

Dear Dr. Pajogo,

I'm pleased to inform you that your manuscript has been deemed suitable for publication in PLOS ONE. Congratulations! Your manuscript is now being handed over to our production team.

Kind regards,

on behalf of

Dr. Laura Hannah Kelly

Staff Editor

PLOS ONE